# Photoswitchable Molecular Units with Tunable Nonlinear Optical Activity: A Theoretical Investigation

**DOI:** 10.3390/molecules28155646

**Published:** 2023-07-26

**Authors:** Aggelos Avramopoulos, Heribert Reis, Demeter Tzeli, Robert Zaleśny, Manthos G. Papadopoulos

**Affiliations:** 1Department of Physics, University of Thessaly, 35100 Lamia, Greece; 2Institute of Chemical Biology, National Hellenic Research Foundation, 11635 Athens, Greece; hreis@eie.gr (H.R.); mpapad@eie.gr (M.G.P.); 3Laboratory of Physical Chemistry, Department of Chemistry, National and Kapodistrian University of Athens, 15784 Athens, Greece; tzeli@chem.uoa.gr; 4Theoretical and Physical Chemistry Institute, National Hellenic Research Foundation, 11635 Athens, Greece; 5Faculty of Chemistry, Wrocław University of Science and Technology, Wyb. Wyspiańskiego 27, PL-50370 Wrocław, Poland; robert.zalesny@pwr.edu.pl

**Keywords:** (hyper)polarizability, density functional theory, molecular switches, photochromism, two-photon absorption, dithienylethene, bis(ethylene-1,2-dithiolato), excited-state energy transfer

## Abstract

The first-, second-, and third-order molecular nonlinear optical properties, including two-photon absorption of a series of derivatives, involving two dithienylethene (DTE) groups connected by several molecular linkers (bis(ethylene-1,2-dithiolato)Ni- (NiBDT), naphthalene, quasilinear oligothiophene chains), are investigated by employing density functional theory (DFT). These properties can be efficiently controlled by DTE switches, in connection with light of appropriate frequency. NiBDT, as a linker, is associated with a greater contrast, in comparison to naphthalene, between the first and second hyperpolarizabilities of the “open–open” and the “closed–closed” isomers. This is explained by invoking the low-lying excited states of NiBDT. It is shown that the second hyperpolarizability can be used as an index, which follows the structural changes induced by photochromism. Assuming a Förster type transfer mechanism, the intramolecular excited-state energy transfer (EET) mechanism is studied. Two important parameters related to this are computed: the electronic coupling (V_DA_) between the donor and acceptor fragments as well as the overlap between the absorption and emission spectra of the donor and acceptor groups. NiBDT as a linker is associated with a low electronic coupling, V_DA_, value. We found that V_DA_ is affected by molecular geometry. Our results predict that the linker strongly influences the communication between the open–closed DTE groups. The sensitivity of the molecular nonlinear optical properties could assist with identification of molecular isomers.

## 1. Introduction

Multi-photochromic molecules are of current interest, since such systems allow access up to 2^n^ molecular states, if *n* photochromic units are distinguishable, while molecules involving one photochromic unit are bistable [1,2]. For example, a derivative that involves two similar photochromic switches may have the following states: “*open–open*” (**oo**), “*open–closed*” (**oc**), “*closed–closed*” (**cc**). However, in several cases involving DTE switches, the “*closed–closed*” isomer is not observed. This has been attributed to excitation energy transfer (EET) as an efficient competition process for the photoinduced cyclisation. The photo-activity of the involved photochromic groups depends on the linker or molecular bridge, which connects them [2]. It is understood that a saturated linker may allow the photochromic groups to retain their photo-activity.

EET is a process of great importance in many areas of research (e.g., biological systems and opto-electronic devices) [3]. Of particular relevance to the present work, and the most frequently observed EET, is singlet excitation energy transfer (SEET). SEET has been used to rationalize several important processes (e.g., light-harvesting in photosynthesis) [4].

An important parameter for the understanding of EET is the electronic coupling factor between the two switching units; this is an off-diagonal Hamiltonian matrix element between the initial and final diabatic states in the transfer processes [3]. This coupling involves a Coulomb and a short-range term [3]. At large separations, the former reduces to the Förster dipole–dipole coupling [5]. The latter involves Dexter’s exchange coupling [6] and an overlap term [7]. Furthermore, another term related to the effect of the solvent may also play a role [8].

Overlap of the donor fluorescence spectrum with the acceptor absorption spectrum ensures the energy conservation of the EET process in the weak coupling limit [9]. Kaieda et al. [10] studied the photocyclization of dithienylethene multimers and reported that the overlap of the fluorescence spectrum of the (**oo**) dimer and the absorption spectrum of the (**oc**) form suggest that intramolecular energy transfer, from the excited open-ring fragment to the closed-ring unit, is possible.

The significant role of the bridge on intramolecular EET has been studied by several authors; in particular, Chen et al. [11] investigated the effect of the bridge on the Coulomb coupling, which makes a major contribution to electronic coupling. They found that the EET rate increases, in comparison to through-space models, when the donor and acceptor transition dipoles are arranged longitudinally and are linked by a polarizable bridge.

Scholes et al. [12] studied the through-space and through-bond effects on exciton interactions in a series of dinaphthyl molecules, in which the naphthyl units are connected by a polynorbornyl bridge. The enhancement of the energy delocalization by the through-bond interactions has been noted. Electronic coupling through rigid saturated spacers, involving up to 12 ***σ*** bonds, has been reported [12,13]. 

Energy transfer rates have been studied in donor–bridge–acceptor (D–B–A) systems [14]; several factors have been considered, e.g., the bridge length [15,16], conformation [17], and electronic properties [18]. McConnell’s super-exchange theory for electron transfer [19] has also been frequently applied to energy transfer [14]. 

Kudernac et al. [20], in their study of uni- and bi-directional light-induced switching of DTEs, linked to the surface of gold nanoparticles, found that the ring-closure process depends on the spacer. 

If several DTE units are present in a molecule, a *stepwise* photocyclization process is significant. Non-conjugated linkers are associated with a fully ring-closed isomer [21], because no SEET takes place. However, if π-conjugated linkers are involved, a *partial* photochromism is usually observed, due to SEET. 

Perhaps the simplest approach to compute EET is using Förster’s theory [22]. This approach has been successfully applied in many cases (e.g., in predicting EET rates) [22]; however, there are several cases where it has failed [23,24]. This method assumes that there is a very weak coupling between an open DTE unit (D) and a closed one (A) and that their spatial extensions are much smaller than R_DA_ (the distance between D and A), so that the point–dipole approximation can be employed [22]. Several methods have been proposed for a more accurate description of EET [3,24,25,26,27]).

Calculation of the electronic coupling between chromophores is to approximate it using the Coulomb interaction and to compute it by employing the transition dipole moments, that is, the point–dipole approximation (PDA), which is an easy way to calculate the coupling, but it fails at short distances [28]. One may go beyond the dipole–dipole approximation to involve multipole couplings. For short distances, these have been found to have significant contributions [29]. A less approximate approach to compute the Coulomb coupling is to use the density cube method [27]. The point–dipole approximation (PDA) is frequently used, giving satisfactory results, when the distance between the donor and the acceptor (R_DA_) is much larger than their dimensions. Overestimated results may be received at short distances [23].

The study of molecular optical properties provides valuable information in several important areas (e.g., nano-structures) [30]. Studies of single molecules help us to probe and to understand mechanisms in deeper detail and could be used to obtain information on what is happening at the nanoscale [30]. The third-order nonlinear optical response of energy transfer systems has been studied theoretically by Young and Fleming [31]. The effect of molecular switching on the NLO properties has also been studied by several research groups [32,33,34]. The stimuli (e.g., light irradiation and pH variation), which were used to induce the molecular transformation, have also been discussed [33]. The DTE switches, in connection with light of appropriate frequency (which leads to “*open*” or “closed” DTE units) allow efficient control of the molecular linear and nonlinear optical (L&NLO) properties, since the cyclization reaction increases the conjugation, and thus polarization DTE multistate molecular materials have been considered as a research field of significant interest [35,36]. In addition, EET is a significant property of excited states and provides an important tool for the rationalization of the L&NLO properties. Thus, the photochromic properties of DTEs, their L&NLO, and EET provide complementary information about molecular structure. 

In view of the challenging issues mentioned, we here address the following topics:(i)The relationship between the molecular (hyper)polarizabilities of several derivatives consisting of two DTE units, with and without different substituents, and connected by different conjugated linkers, and the changes induced by photochromism on these properties. It is a major objective to find those structures and linkers, in particular, that lead to a significant contrast between the hyperpolarizabilities of the “*open*” and the “*closed*” isomers.(ii)The effect of the linker on the electronic communication of the DTE units and the photochromism as well as the intramolecular EET. A series of linkers has been selected in order to tune EET and eventually to minimize it, to attain full photochromism.

## 2. Results and Discussion

As mentioned in the introduction, the linker is of crucial importance for the communication between the open–closed DTE units and for the modulation of the molecular NLO properties. Three types of molecular bridges were selected: (a) a quasilinear tetrathiophene chain (see Figure 1), which is known for its remarkable electronic and optoelectronic properties [37]; (b) NiBDT, which provides an excellent basis for the formation of molecular materials with exceptional optical, electronic, and conductive properties [38]; and finally naphthalene, which involves two fused benzene rings; this is a 10 π electron system similar to NiBDT. It should also be noted that two derivatives of molecule **1** (Figure 1), with R = Cl and R = Phenyl, were synthesized and their photochromism studied, thus allowing comparison of the predicted ring closure of both DTE groups with the corresponding experimental findings. A 1D π-conjugated NiBDT nanosheet has also been synthesized [39].

The main reasons for selecting the above bridges are, first, the expected significant contrast in the NLO properties of the three isomers (**oo**, **oc**, and **cc**) and second, the interesting and challenging properties associated with the low-lying excited states of NiBDT, in connection with EET. It should also be noted that each molecular structure (**cc**, **co**, and **cc**) is studied separately, in its lowest energy conformation, since we are interested in probing and understanding the EET–NLO relationship activity mechanism in more detail. Eventually, we want to understand how to attain full photochromism by modifying molecular structure.

### 2.1. (Hyper)polarizabilities

The photochemical process leads to a large change of the structure, which is accompanied by the very structure-sensitive first and second hyperpolarizability properties (Table 1 and Table 2). 


**A. Oligothiophenes**


A series of derivatives with two DTE groups connected by a tetrathiophene group have been studied (Table 1 and Figure 1), with different substituents R/R: H/H, Cl/Cl, NO_2_/NO_2_, NH_2_/NH_2_, Ph/Ph, and NO_2_/NH_2_ on the DTE groups. We note that these derivatives may also be considered as alkene, methyl end-capped sexithiophenes, which may be a more adequate characterization at least for the open–open isomers [37].

**Polarizabilities.** The average polarizabilities of the “open–open” (**oo**) isomers are: 879 ± 82 a.u. (Table 1), where the limits denote the maximum and minimum average polarizability values among the oo isomers. For the “open–closed” (**oc**) and “closed–closed” (**cc**) isomers the corresponding values are 989 ± 107 a.u. and 1106 ± 133 a.u., respectively. The observed trend is α(**cc**) > α(**oc**) > α(**oo**); this is explained by the increased conjugation associated with the closed DTE thus increasing the electron mobility. For R = Ph, as expected, we observe the larger polarizability.

**First hyperpolarizabilities**. The largest β value is observed for the pair R,R’: NO_2_/NH_2_ of **1cc** (71,350 a.u.) and **1co** (61,250 a.u.) isomers (Figure 1). This pair of substituents gives very different first hyperpolarizability values depending heavily on the state of the DTE unit (closed or open) to which it is bonded. For the **1co** isomer, the larger β value is observed when NO_2_ is bonded to the closed DTE unit and NH_2_ is bonded to the open one. The tetrathiophene unit is conjugated (Figure 1). We shall now consider the effect of the extension of the conjugation path due to the closed DTE unit(s). The effect of extending the conjugation path on β may be seen when R: H/H (Table 1); the maximum β value (14,860 a.u.) is observed for **1co**. 

The impact of extending the conjugation path on the first hyperpolarizability may also be seen when R: Cl/Cl, NO_2_/NO_2_, NH_2_/NH_2_, or Ph/Ph, but in these cases *charge transfer* may also affect the results. The effect of charge-transfer on β, may be seen by the difference:Δ1 = β(Is; R:NO_2_/NH_2_) – (β(Is; R:NO_2_/NH_2_)+ β(Is; R:NH_2_/NO_2_))/2(1)
where Is: **1cc**, **1oc**, or **1oo**. 

This difference takes the values 70,956 a.u., −21,270 a.u., 37,040 a.u., and 4321 a.u. for **1cc**, **1co**, **1oc**, or **1oo**, respectively (Table 1). The effect of charge transfer, as a function of conjugation, upon photoswitching*,* may be expressed using the above sequence of values. We observe that the molecular state, the conjugation, and the charge transfer have a very significant effect on the first hyperpolarizability (β).

**Second hyperpolarizabilities**. Larger property values are observed for the cc isomer, as would be expected due to the conjugation; the following trend is observed: γ(**cc**) > γ(**oc**) > γ(**oo**) (Table 1). In particular, we note that the effect of extending the conjugation path on γ may be seen via comparison with R,R’:H/H; the maximum second hyperpolarizability value is observed for **1cc** (14,111 × 10^3^ a.u.). The effect of *charge transfer* on γ may be estimated using the difference:Δ2 = γ(Is; R,R’:NO_2_/NH_2_) – (γ(Is; R,R’:NH_2_/NO_2_) + γ(Is; R,R’:NO_2_/NH_2_))/2(2)
where Is: **1cc, 1co, 1oc**, or **1oo**. This relationship can be shown to be approximately valid in the two-state approximation to γ [40], if the average transition dipole of the two symmetrically substituted compounds is similar to the transition moment of the unsymmetrically substituted one, and the transition frequencies are also similar.

This difference takes the values 5844 × 10^3^ a.u., −2056 × 10^3^ a.u., 3757 × 10^3^ a.u., and 26 × 10^3^ a.u. for **1cc**, **1co**, **1oc**, and **1oo**, respectively (Table 1). Again, the effect of charge transfer, as a function of the conjugation length, may be expressed using the above sequence of values. We observe that both conjugation enhancement, upon photoswitching, and charge transfer have a very significant effect on the second hyperpolarizability (γ).

Taking the ratios Δ1(**1cc**)/Δ1(**1oo**) = 16.9 (Equation (1) and Δ2(**1cc**)/Δ2(**1oo**) = 224.8 (Equation (2)), we observe that the effect of *charge transfer*, as a function of conjugation, has a much greater effect on γ than on β. The corresponding ratio for the polarizability (α) is 3.4. In general, α obeys most of the trends of γ, but in a less pronounced way. Finally, let us also note that for the case of α and γ, change of the geometry (B3LYP/M062-X) has s small effect on the contrast ratios, k = P**_1cc_**/P**_1oo_** and λ = P**_1cc_**/P_1co_ (Table 1; R = Cl). It is observed that k = 1.24/1.17(α) and 3.74/3.05 (γ), while λ = 1.11/1.11 (α) and 1.68/1.83(γ). The notation A/B(P) stands for the ratio computed at the B3LYP/M062-X-optimized geometry, and P = α, γ denotes the property. For the case of β, the effect was found to be larger and stands for k = 4.9/13.6 and λ = 0.06/0.17.


**B. Derivatives Having NiBDT and Naphthalene as Linkers**


A series of derivatives were studied, which involve two DTE groups, connected by either NiBDT or naphthalene (Figure 2 and Table 2). Both linkers have 10π electrons.

***NiBDT.*** We observe that the cc isomer lies lower in total energy compared with the **oc** and **oo** ones (Appendix A):E(**2oo**) > E(**2oc**) > E(**2cc**)

We also observe that an increase of conjugation, upon photoswitching, leads to a decrease of the |HOMO-LUMO| (=Δ_HL_) gap (Appendix A), due to destabilization of HOMO; for **2oo**, **2oc**, and **2cc**, the corresponding Δ_HL_ values are 0.059 a.u., 0.050 a.u., and 0.042 a.u., respectively. Let us note that a similar trend was found for the oligothiophene derivatives (Appendix A).

These results are in agreement with those reported in the literature [41]. A red shift of λ_max_ is also noted, upon changing the structure of the isomer: λ(**2cc**) > λ(**2oc**) > λ(**2oo**) (Table 2). 

For the (hyper)polarizabilities, we observe the trend (Table 2): P(**2cc**) > P(**2oc**) > P(**2oo**), where P: α, |β|, and γ. The cc isomer has remarkably larger values for the properties (α, |β|, and γ)than the other isomers.

**Naphthalene.** For the total energy, we also observe that: E(**3oo**) > E(**3cc**) (Table 2). A similar trend was found for NiBDT as a linker. For the (hyper)polarizabilities, we found: P(**cc**) > P(**oo**), P = α,β,γ. A significant change for β and γ of **oo** and **cc** isomers is also observed. 

NiBDT and naphthalene both have 10π electrons; however, their effects on α, β, and γ are very different. For NiBDT, as a bridge, the ratio P**_2cc_**/P**_2oo_**, where P= α, β, or γ is 1.8, 34.3 and 24.7, respectively, while for naphthalene the corresponding ratios for **3cc**/**3oo** are 1.42(α), 13.4(β), and 9.9(γ). The ratio P**_2cc_**/P**_3cc_**, which shows the effect of the type of bridge on the L&NLO molecular properties, is 1.4, −2.4, and 4.9 for α, β, and γ, respectively. As it has been shown, the existence of low-lying excited states, due to the presence of NiBDT, significantly enhances the polarization character [42]. 

In a previous article [43] we investigated the contrast between the L&NLO properties of the “open” and “closed” isomers of the derivatives involving one DTE unit [43]. We found that P(**c**) > P(**o**), where P: α or γ. In this work, the considered molecules involve two DTE groups. We observe, in general, P(**cc**) > P(**co**) > P(**oo**), where P: α or γ. This trend is due to an increase of conjugation, upon photoswitching, in the order **cc** > **oc** > **oo**. 

As shown in Ref. [43], increasing the conjugation path increases the positive contribution of the density of the second hyperpolarizability much more than the negative one, thus reinforcing the γ values. The trends observed in the first hyperpolarizability are less regular. 

Summarizing the findings from Table 1 and Table 2, we note that the molecular L&NLO optical properties follow the structural changes induced by photochromism. The first hyperpolarizability (β) shows a less regular dependence in comparison to polarizability (α) and second hyperpolarizability (γ). There is a significant contrast between the computed values, for the first and second hyperpolarizabilities of the open–open and closed–closed isomers of the considered derivatives. NiBDT as a linker leads to a greater contrast, in comparison to naphthalene and oligothiophene. The great effect of conjugation on the (hyper)polarizabilities is clearly demonstrated by the closed–closed isomers. The results (Table 1 and Table 2) show that the (hyper)polarizabilities, and in particular γ, can be used as an index, which follows the structural changes induced by photochromism. Therefore, the sensitivity of the polarization character upon photoswitching can assist in the identification of the molecular isomers (**oo-oc-cc**) upon light irradiation. To reinforce the validity of the basis set 6-31G* for the properties considered here, we have computed selected cases with the larger basis set cc-pVTZ. As shown in Table 1 and Table 2, there is reasonable agreement between the two sets of data.

**HOMA analysis.** To obtain further insight into the reasons for the differences of the (hyper)polarizabilities for the different isomers, we computed a measure for the conjugation along the quasilinear conjugated [-C = C-]_n_ chain, following the prescription of the Harmonic Oscillator Model of Aromaticity (HOMA) index, *I*_HOMA_ [44]:(3)$${\;I}_{HOMA}=1-\frac{\alpha }{n}\sum_{i=1}^{n}{\left(R_{i}-R_{Opt}\right)}^{2}$$
where *n* is the number of CC bonds; *R_i_* is the *i*-th bond length of the conjugated chain; *R_Opt_* is the reference bond length in benzene, chosen as an optimally conjugated system and computed at the same level of theory (1.397 Å at B3LYP/6-31G*); and 257.7 Å^−2^ is a normalization factor chosen such that *I*_HOMA_ of an aromatic compound approaches 1 and that of its Kekulé non-aromatic structure becomes 0.

The HOMA index was used to compute two of the R_1_-1xy-R_2_ series of isomers (x,y = c,o): (a) with R_1_ = NO_2,_ R_2_ = NH_2_ and (b) with R_1_ = R_2_ = H. For the internal [-C = C-]_21_ chain, which can be written as a neutral, fully conjugated pseudolinear structure in all isomers, the calculated HOMA indices using Equation (3), with n = 21, are shown in Table 3. For both molecules, the conjugation as measured using the HOMA index increases with the number of closed-ring structures.

### 2.2. Two-Photon Absorption

The electronic structure parameters (excitation energies, oscillator strengths, and two-photon transition strengths) corresponding to five lowest-energy electronic excitations of **1cc, 1co,** and **1oo** molecules (Figure 1, R = H) are shown in Appendix A. Note that the values of two-photon absorption (2PA) strengths for higher-lying states (S_4_–S_5_) are not very accurate due to resonance effects (they are overestimated). In order to obtain more realistic values, one should employ damped response theory [45], but given the size of the studied systems such calculations were not feasible. We will start with the analysis of one-photon absorption properties. The data shown in Appendix A allow us to make a few general comments. 

First, for all considered substituents, the S_0_ → S_1_ transition is characterized by very large values of oscillator strengths (spanning the range 1.18 (R = NO_2_, **oc**) – 2.54 (R = NO_2_, **oo**)). 

Second, there is a common pattern for all compounds for this transition: ∆E(**cc**) < ∆E(**oc**) < ∆E(**oo**)
and
f(**oc**) < f(**cc**) < f(**oo**) 

Third, for **cc** and **oc** isomers with R = NO_2_ and R = NO_2_, NH_2_ substituents, there is a moderate change in S_0_ → S_1_ excitation energy in comparison to other derivatives (R = Cl, R = H, R = NH_2_). However, for **oo** we note that the S_0_ → S_1_ excitation energy is insensitive to the change of substituent, i.e., it is roughly 2.8 eV. The 2PA properties corresponding to the S_0_ → S_1_ transition are negligible (i.e., less than 10^4^ au) for most of the compounds. There are only three exceptions, i.e., **oc** isomer (R = NO_2_; Appendix A) and **cc** and **oc** with R = NO_2_, R′ = NH_2_ (Appendix A). The corresponding values for all three cases exceed 5 × 10^4^ and reach up to 12 × 10^4^ (**cc** isomer). Taken together, these results demonstrate that in the case of bright S_0_ → S_1_ transition it is possible to tune the excitation energies and 2PA transition strengths by changing substituents only for **cc** and **oc** isomers. The most pronounced change in properties is achieved using asymmetric substitution of NO_2_ and NH_2_, and 2PA strengths can be increased by two orders of magnitude.

### 2.3. Excitation Energy Transfer

The intra-molecular EET and the overlap between absorption and emission spectra of the studied systems will be analyzed below. The intra-molecular process depends on a number of factors (e.g., V_DA_—the electronic coupling between the donor and the acceptor, the overlap of the emission spectrum of the donor, and the absorption spectrum of the acceptor). By necessity, this study has to be selective. Thus, analysis of the intra-molecular EET will rely on V_DA_. Additionally, the overlap of the emission spectrum of the donor and the absorption spectrum of the acceptor will be studied. 

The selected bridges, between the DTE units (Figure 1, Figure 2 and Figure 3), allow tuning of V_DA_ and eventually EET. In this context, it is useful to find out how the modification of the structure affects V_DA_. In particular, it is important to specify rules to minimize EET and thus to attain full photochromism. This is a central question of our study. 

#### 2.3.1. Intra-Molecular Excited Energy Transfer

For the study of the intra-molecular EET (excited energy transfer) we have employed the following approaches:

(i) Two methods have been used for the computation, for V_DA_ of several models, including coupling between the donor (open unit) and the acceptor (closed unit). 

(ii) For the case of **1oc** isomer, the emission spectrum of the donor and the absorption spectrum of the acceptor have been analyzed.

In the following section we will discuss the results, obtained by the two methods we have used to calculate V_DA_. 

**Linear response method**. The ground-state equilibrium geometries have been used to compute V_DA_ [1]. We employed three types of molecular bridges, NiBDT, naphthalene, and an oligothiophene (Figure 1, Figure 2 and Figure 3). For NiBDT and naphthalene, the cis and trans isomers of the corresponding derivatives were considered. For the trans isomer of naphthalene, as a bridge, we used two conformations; one is planar and in the other, the plane of naphthalene is vertical to the plane of the DTE groups. NiBDT is bonded with the two switches with two different ways: (i) NiBDT is fused with the two DTE units (**2oc**; Figure 2), and (ii) it is bonded with the switches with single bonds (**7oc** and **8oc**; Figure 3). Each molecule is decomposed into two fragments: D (donor; the open DTE) and A (acceptor; the closed DTE bonded with the bridge). The bond, which is cut, is capped with a hydrogen atom at both ends. Thus, a well-defined fragment results. The employed models are given in Figure 3. The linker (or bridge), i.e., naphthalene, NiBDT, or oligothiophene, is bonded to the closed DTE unit (acceptor). This model corresponds to the M2 model employed by Fihey et al. [1]. It has been selected because it was found in Ref. [1] the when the bridge is bonded to the acceptor unit, the Coulomb interaction is enhanced, and thus a significant EET process takes place. A detailed discussion of this and several other models is given in the above reference. For the computation of V_DA_, one needs to find the excited states, which participate in the EET process [1]. For the open DTE moiety, which is the donor fragment, the excited state involves, in most cases, a HOMO-LUMO transition; this is connected with the photocyclization process. Thus, the selected relevant transition energy should lie in the UV region. The relevant excited state of the closed DTE unit, which is the acceptor fragment, is selected by taking into account two criteria: (i) its energy should be close to or lower than the energy of the donor, and (ii) it should have non-negligible oscillator strength. It has been shown, based on Förster’s theory, that a significant suppression of the EET rate takes place when the excitation energy difference between the donor and the acceptor is large, in comparison with other relevant energy properties of the interacting molecules [46]. Table 4 reports two contributions to V_DA_: the coulombic (V_c_) and the exchange (V_xc_).

Areephong et al. [37] observed ring closure of both photochromic units for the sexithiophenes they studied (**1oc**; Figure 1; R: Cl). In agreement with experimental observation, a low value of V_DA_ was computed for R = Cl (8.3 cm^−1^; Table 4), suggesting ring closure. In the second derivative considered (**2oc**; Figure 2), each DTE group is fused with two NiBDT molecules, while the Ni compound also operates as a bridge. A rather low V_DA_ has been computed for this (10.3 cm^−1^). Therefore, if we consider this value of V_DA_ (8.3 cm^−1^) as a reference, it is likely that a NiBDT bridge allows switching between the three states, although for a more complete answer, other factors should also be considered (e.g., energy transfer speed and cyclisation time) [1].

We observe that for **2oc**, the relevant excited state of the donor is the 22nd (268 nm; Appendix A). This is, of course, quite high and reinforces the conjecture of a very low probability of EET. Nevertheless, although it is known that the higher-energy emission bands in the UV would violate Kasha’s rule and the Kasha−Vavilov rule for fluorescence, violations of these rules are reported specifically in connection with intramolecular EET [47,48,49,50]. Figure 4 presents the emission spectrum of the fragment, which operates as a donor; the absorption spectrum of the fragment, which acts as acceptor; and their overlap. The emission, which is relevant for EET, is found in the area of 230–300 nm. We observe two peaks at 250 nm and 270 nm. However, two antagonistic processes are likely to take place before EET proceeds: internal conversion to S_1_ and photocyclization of the open DTE group. In addition, all the oscillator strength (*f*) values associated with the emission of UV, except that of the 22nd state, are negligible (Appendix A). Taking into account these considerations and the corresponding small value of V_AD_ (10.3 cm^−1^), we believe that EET is unlikely to take place, and thus the open DTE unit of **2oc** is likely to close.

Dimers **4oc**, **5oc**, and **6oc** have naphthalene as a linker; **4oc** involves the DTE groups in cis orientation, while **5oc** and **6oc** have the DTE units in trans orientations. In **5oc** and **6oc**, the naphthalene groups have a different arrangement (Figure 3). In all considered cases, V_xc_ is very small. We observe that V_DA_ is very large for **4oc**–**6oc**, where naphthalene is the linker. This suggests significant EET and, thus, partial photochromic activity (i.e., the closing of only one photochromic unit is likely to be observed). The significant effect of geometry on V_DA_ is clearly demonstrated by these results (Appendix A).

Dimers **7oc** and **8oc** involve NiBDT as a linker. Dimers **7oc** and **8oc** have the DTE groups in cis and trans orientation, respectively. We observe that for **7oc** and **8oc**, the excited states, S_1_–S_3_, of the donor have for emission *f* = 0. Thus, S_4_ was considered. The smaller V_DA_ is observed for the trans isomer, **8oc** (12.8cm^−1^). This low V_DA_ suggests that no significant EET takes place; therefore, both DTE groups are likely to close.

It is interesting to compare the V_DA_ (Coulomb contribution, V_C_) value of naphthalene as a linker with that of a biphenyl group as a bridge. In the first case, V_C_ = 101.6–120.9 cm^−1^, depending on the orientation of the open/closed units of the linker; in naphthalene, the two phenyl groups are fused. In the second case, V_DA_ = 3.1cm^−1^; the two phenyl groups are connected with a single bond. The great effect of the structure on V_DA_ is clearly seen.

We observe that for **5oc** (Appendix A and Figure 3), the transition energy for the acceptor, 4.209 eV, is associated with the fourth excited state, S_4_. However, for **8oc**, the corresponding transition energy 4.136 eV is associated with the excited state, S_50_. This very large number of intervening excited states is due to the presence of NiBDT [42]. The unusually large NLO properties of NiBDT derivatives (Table 2) are also due to the significant number of low-lying excited states. 

The calculation of V_DA_, using the linear response method, illuminates some aspects of EET, that is, the excitation of the open DTE and the absorption of the closed DTE, which are due to the de-excitation of the open DTE.

Our results clearly show that the linker strongly influences the communication between the DTE groups and thus, photochromism. The molecular geometry has a significant effect on V_DA_. NiBDT, as a linker, is associated with relatively low V_DA_ values; therefore, both DTE groups are likely to close. This low V_DA_ is due to the near-IR absorption spectrum of NiBDT. Our computations show that the considered oligothiophene, as a linker, is associated with low V_DA_ and thus allows both DTE groups to close. This is in agreement with experimental observation [37].

**Distributed multipole analysis (DMA).** To gain a further understanding of the EET process, we have also computed the V_c_ contribution of V_DA_ using the method proposed by Błasiak et al. [51], which relies on the distributed multipole expansion of the transition densities. The results for the DMA treatment, as a function of the limiting value of the sum of the interacting multipole moments *l* + *l′* (where *l*,*l′* = 0 (charge), …, 4 (hexadekapole)), are given in Table 5. 

It has been shown in Ref. [51], and was also found here, that limiting the sum of the ranks yields better results than the alternative approach of using all multipoles up to a limiting rank *l*. As the values show, the electrostatic contribution computed using the multipole expanded transition density employing the highest limiting rank l + l’ = 4 compare quite well with the analytically computed values (linear response method); the differences are in the range of 10^−4^ eV (~0.8cm^−1^). We note that both the added terminal hydrogen H_term_ as well as the carbon atom connected with it were not used as explicit expansion centers in the DMA treatment. The remaining differences between the multipole-expanded method and the analytical values may be caused at least in part by a not-yet-converged expansion series and/or by the charge density of H_term_, which has not been removed. Although H_term_ was not used as an expansion center, its associated charge density is still taken into account in the DMA treatment. 

We concluded that the results for V_c_, computed using the DMA method, are in satisfactory agreement with those calculated using the linear response approach (Table 4 and Table 5).

#### 2.3.2. Overlap of the Emission Spectrum of the Donor and the Absorption Spectrum of the Acceptor

In this section we shall discuss the overlap of the emission spectrum of the donor and the absorption spectrum of the acceptor (Figure 4). The considered systems involve **2oo**, **2oc**, and **2cc** (Figure 2), where for the pair **2oo/2oc**, **2oo** is the donor, and **2oc** is the acceptor, i.e., **2oo** absorbs photons and then emits, while **2oc** absorbs the emitted photons. Similarly, for the pair **2oc/2cc**, **2oc** is the donor, and **2cc** is the acceptor, i.e., **2oc** absorbs photons and then emits, while **2cc** absorbs the emitted photons. This overlap is an indication that EET phenomena may occur between different molecules. Of particular relevance for this work is overlap in the UV area, because photocyclization takes place at this area. One of our major themes is whether EET will allow the open DTE unit of **2oc** to close, leading to **2cc**.

For the case of the **2oo** structure, two conformers **a** and **b** have been obtained. Rotations around the C–C bond convert the **a** isomer to **b** (see Figure 5). 

The **a** conformer is lower in energy than **b** by 0.18 eV because weak π–π interactions are formed between the rings of the central NiBDT and the rings of the other NiBDT groups. However, both present similar absorption spectra, i.e., their peaks differ by less than 6 nm and up to 0.03 eV (see below). Finally, it should be mentioned that the optimized geometry of the excited states of **2oo**, that corresponds to the main peaks of the emission spectra, has similar geometry to **b**.

The absorption spectrum of **2oo** has a main peak in the visible range at ~600 nm, an intense peak in UV area at ~310 nm, and small overlapping peaks at ~700 nm and ~900 nm in the NIR area. On the contrary, the main peaks of the absorption spectra of **2oc** and **2cc** are found at 807 and 862 nm in the NIR area. At 650 and 600 nm, there are some overlapping peaks for **2oc**, and at 670 nm for **2cc**. Finally, both present small peaks in the UV area of 400–300 nm (see Figure 6, Appendix A, and Table 6 and Appendix A). The emission spectrum of **2oo** presents two peaks of similar intensity at 670 nm and 313 nm in the vis and UV regions, respectively. The emission spectrum of **1oc** presents three peaks: an intent at 917 nm in the NIR region and two peaks of similar intensity at 662 nm and 326 nm in the vis and UV areas.

The vertical and adiabatic excitation energies with respect to the ground state and vertical emission de-excitation are shown in Table 7. We found that the absorption spectra of **2oo** presents two main peaks that correspond to excitations of 2.07 and 4.01 eV and a small peak that corresponds to excitations of 1.81. As new rings are formed (**2oo → 2oc → 2cc**), less energy is needed for these three vertical excitations, i.e., the excitations are red-shifted for the **2oc** structure up to 0.3 eV and further red-shifted for **2cc**. The most intense, red-shifted excitation is for the small peak at 1.81 eV, and the least red-shifted excitation is for the main peak of the 4.01 eV. Similarly, the vertical emission and the adiabatic excitation are similar or less for **2oc** compared to **2oo**. 

It is of interest that the vertical main emitting peaks of **2oo** are at 1.44, 1.85, and 3.97 eV, very similar to the absorption peaks of **2oc**, which are at 1.54, 1.87, and 3.92 eV (see Table 7). This shows that energy transfer is possible from the excited open-ring unit to the closed-ring unit, i.e., **2oo → 2oc**. Similarly, the vertical main emitting peaks of **2oc** are at 1.35, 1.86, and 3.81 eV, and the adiabatic de-excitation energies are at 1.40, 1.87, and 3.88 eV; thus, they are very similar to the absorption peaks of **2cc**, which are at 1.44, 1.84, and 3.92 eV. Again, energy transfer is possible from the excited open-ring (**2oc → 2cc**) unit to the closed-ring unit. Conclusively, energy transfer is possible through this **2oo → 2oc → 2cc** process.

The findings of the reported excited state computations are in agreement with the conclusions presented in Ref. [38]. The authors there reported that excited-state quenching is responsible for the absence of photochemistry in dithienylethene/oligothiophene-based polymer while intramolecular quenching is observed for the dithienylethene units by the sexithiophene unit.

Finally, the frontier molecular orbitals (MO) involving in the main excitations are depicted in Figure 7, Figure 8 and Figure 9. It is of interest that many MO orbitals are involved in the main excitations (see Appendix A and Table 6). The main absorption peaks of the **a** conformer in the vis and UV areas are at 599 nm and 310 nm, and they present a small charge-transfer (CT) character. This results from the fact that in the **a** conformer π–π interactions exist, which are observed in the occupied orbitals (see for instance HOMO and HOMO-6 in Figure 7). In the unoccupied LUMO and LUMO+2 orbitals, where the electron is transferred via excitation, it is in the peripheral NiBDT group. On the contrary, in the **b** conformer no CT character is observed due to the absence of the π–π interactions. The fluorescence peaks of **2oo** at 859 nm and 670 nm do not have a CT character; however, the peak at 312 nm does (see Figure 7). The three peaks of the absorption spectrum of **2oc** at 807, 664, and 316 nm have only a small, partial CT character. The emitting peak at 326 nm is a CT de-excitation from d orbitals of Ni to p electrons of S (see Figure 8). Finally, regarding **1cc**, where additional aromatic rings are formed, the absorption peak at 317 nm has a clear CT character, with a significant f coefficient of 0.14, from the peripheral C_2_H_2_S_2_Ni group to the whole molecule, i.e., the electron density is delocalized in all **2cc** structures (see Figure 9).

## 3. Methods

We shall discuss the following topics in this section: (i) a definition of the hyperpolarizabilities; (ii) the functionals we have employed; (iii) the validation procedures we have used; (iv) the method used to compute the transition energies; (v) the computational approach employed to calculate the emission spectrum of the donor, the absorption spectrum of the acceptor, and their overlap; (vi) the procedure used to compute two-photon absorption; and (vii) the methods employed for the computation of the electronic coupling.

### 3.1. Definitions

A Taylor series may be used for the expansion of the energy (E) of a molecule, which is placed in a static, uniform electric field (F_i_) [52]:(4)$$E\left(F\right)=E^{0}-\mu_{i}F_{i}-\left(\frac{1}{2}\right)a_{ij}F_{i}F_{j}-\left(\frac{1}{6}\right)\beta_{\iota jk}F_{i}F_{j}F_{k}-\left(\frac{1}{24}\right)\gamma_{\iota jkl}F_{i}F_{j}F_{k}F_{l}-\cdots$$
where E^0^ is the field free energy, and $\mu_{i}$, $a_{ij}$, $\beta_{\iota jk}$, and $\gamma_{\iota jkl}$ are the dipole moment, polarizability, and first and second hyperpolarizability components, respectively; a summation over repeated indices is implied. The average (hyper)polarizabilities are defined using:(5)$$a=\frac{1}{3}\left(a_{xx}+a_{yy}+a_{zz}\right)$$
(6)$$\beta =\sum_{i=x,y,z}\frac{\mu_{\iota \;\beta_{\iota }}}{\left\|\mu \right\|}$$
where β_ι_
(7)$${\;\beta }_{\iota }=\frac{1}{5}\sum_{j=x,y,z}{(\beta }_{\iota jj}+\beta_{jij}+\beta_{jji})$$
(8)$$\gamma =\frac{1}{5}\left(\gamma_{xxxx}+\gamma_{yyyy}+\gamma_{zzzz}+{2\gamma }_{xxyy}+{2\gamma }_{xxzz}+{2\gamma }_{yyzz}\right)$$

The finite field perturbation theory (FPT) has been used for the calculation of all the necessary tensor components of the static (hyper)polarizabilities defined using Equations (5)–(8) [53]. The Romberg–Rutishauer method [54,55,56] has been employed in order to safeguard the numerical stability of the computed (hyper)polarizability values. The computed static (hyper)polarizability values are expected to be useful for a relative comparison of the NLO properties of the studied photoswitchable compounds, since these are a good approximation of dynamic ones (frequency-dependent) in the off-resonant region [57]. The following field strengths have been used: 2^m^F, m = 1–4, and F = 0.0005 a.u. GAUSSIAN 16 software has been employed for the DFT computations (see below) [58]. 

### 3.2. Functionals and Basis Sets

The B3LYP functional has been used to calculate the structure of the considered compounds. For all the molecular structures, vibrational analysis was performed to verify that a real minimum was found on the potential energy hyper-surface. B3LYP is a well-tested functional [59] that has been used for the determination of the structure of several other NiBDT derivatives [42,43,60,61]. Its satisfactory performance has also been demonstrated in the literature [62,63,64]. In the case of the sexithiophene derivatives, the conjugated skeleton is essentially planar, as shown experimentally in crystal [37], and this conformation is used in our computations. In order to further examine the performance of the B3LYP functional on the oligothiophene structures, we made geometry optimization of **1cc** and **1oo** molecules, R = Cl (Figure 1), by employing the M062-X functional, since the latter has been shown to satisfactorily predict the geometries of π-conjugated systems, due to the appropriate amount (54%) of the included HF exchange [65]. The comparison between the geometries is shown in Appendix A.

The long-range corrected version of B3LYP [66], CAM-B3LYP, has been employed to compute the (hyper)polarizabilities of the considered derivatives. Detailed justification for its use has been reported in Ref. [43]. The CAM-B3LYP functional [66] has also been used for the two-photon absorption calculations (II.6). We have employed the 6-31G* and cc-pVTZ basis set for H, C, O, F, and S atoms [67,68,69,70] and the quasi-relativistic effective core potential ECP28MWB(SDD) for the Ni atoms [71]. 

The adequacy of the CAM-B3LYP/6-31G* approach for the computation of the NiBDT properties is well documented [42,61]. NiBDT derivatives, having a singlet diradical character, should in principle be studied with a multiconfigurational wave function. However, it has been shown that broken-symmetry DFT [(U)DFT] gives satisfactory property values [42,43,72]. 

The electronic structure of the NiBDT derivatives (see Figure 5) have been studied via density functional theory (DFT) and time-dependent DFT (TD-DFT). The geometries of the structures are energetically optimized using the B3LYP [59] and the CAM-B3LYP [66] functionals and the 6-31G* basis set for H, C, O, F, and S atoms and the quasi-relativistic effective core potential ECP28MWB(SDD) for the Ni atoms [71]. The absorption and emission spectra of the studied structures were calculated via the TD-DFT. TD-DFT can predict absorption and emission spectra of molecules in very good agreement with experimental spectra [73,74]. Particularly, the CAM-B3LYP functional has been developed to correct for long-range behavior, and it is regarded as appropriate for the computation of the absorption spectra when charge-transfer states are involved [75]. Four main excited states were energetically optimized. In all cases, the absorption and emission spectra of the studied systems were calculated including up to 130 singlet-spin excited electronic states. All calculations were carried out with the Gaussian16 code. Time-dependent density functional theory (TD-DFT) [76,77] in connection with the CAM-B3LYP functional [66] was used to compute the transition energies. It has been reported that this approach gives satisfactory results [78,79]. All the reported computations were employed in the gas phase.

### 3.3. Two-Photon Absorption

The two-photon absorption (2PA) process is described by the imaginary part of the frequency-dependent second hyperpolarizability [80,81,82]. At the molecular scale, the two-photon absorption process is characterized by the second-order transition moment *S_ab_* that can be computed from the single residue of the quadratic response function [83]. In this work, we assumed one source of photons and linearly polarized light. In such a case, the orientationally averaged two-photon absorption strength for an isotropic medium can be computed as [84]:(9)$$<\delta^{2PA}>=\frac{1}{15}\sum_{ab}\left(S_{aa}\;S_{bb}^{*}+{2S}_{ab}\;S_{ba}^{*}\right)$$
where <δ^2PA^>, in what follows given in atomic units, is directly related to the two-photon absorption cross section (σ^2PA^), which can be determined experimentally. The interested reader is referred to other work for conversion to macroscopic units (cross section is commonly expressed in Goppert-Mayer (GM) units) [85]. The two-photon absorption calculations were performed using the GAMESS US program [86,87] using the CAM-B3LYP functional [66] and 6-31G(d) basis set. The choice of the CAM-B3LYP functional requires a proper justification. In the case of 2PA process, there are several striking reports regarding unsatisfactory performance of exchange–correlation functionals in predicting the magnitude of two-photon strengths [66,85,86,87,88]. For example, the CAM-B3LYP functional gives 2PA strengths that are often underestimated in comparison with the reference coupled-cluster values [85], even though it improves upon conventional functionals in predicting excitation energies to charge-transfer states, as explained above. However, as recently demonstrated for a series of organic molecules, only range-separated functionals (CAM-B3LYP and LC-BLYP) correctly predict changes in δ^2PA^ upon chemical modifications and reproduce experimental trends [89,90]. 

### 3.4. Computation of the Electronic Coupling

The electronic coupling between the excitation of the donor (D) and the acceptor (A), V_DA_, is of major importance for understanding EET; it appears in the energy transfer rate, k_EET_: k_EET_ = (2π/ћ)| V_DA_|^2^J(10)
where J is the spectral overlap (i.e., the overlap integral) between the emission band of the donor and the absorption band of the acceptor [1]. We have computed V_DA_ using two methods: one based on the linear response method and the other based the distributed multipole approach.

**Linear response method**. According to this approach V_DA_ is given by [8]:
(11)$$V_{DA\;}=\int dr\int dr^{\prime }\rho_{D}^{tr*}(r)\frac{1}{\left|r-r^{\prime }\right|}\rho_{A}^{tr}(r^{\prime })+\int dr\int dr^{\prime }\rho_{D}^{tr*}(r)g_{xc}(r,r^{\prime })\rho_{A}^{tr}(r^{\prime })-\omega_{0}\int dr\rho_{D}^{tr}(r)\rho_{A}^{tr}(r)$$
where ρ^tr^ gives the transition density of the donor and the acceptor. The Coulomb interaction between the transition densities is given by the first term, g_xc_ is the exchange–correlation kernel, and the second term gives the exchange–correlation interaction; ω_0_ is the average resonance transition energy of the dimer, while the third term gives a correction contribution.

**Distributed multipole analysis**. For very large molecules, the computational cost of the analytical calculation of the EET terms may be prohibitively large, and a more economical method would be useful. For the Coulomb term, which is generally the largest contribution to the EET coupling, several schemes based on a more sophisticated treatment of the transition densities have been published (see, e.g., the literature cited in Ref. [51]) to explain how to overcome the shortcomings of the original point–dipole treatment using molecular transition dipole moments by Förster [91]. A particularly accurate and computationally economical method, based on the distributed multipole expansion of the transition densities, has been published by Błasiak et al. [51]. In this approach, the electrostatic part of the EET coupling between two excited molecules is computed using transition-density-derived cumulative atomic multipole moments (TrCAMM). Here, we have applied a similar method, replacing TrCAMM with distributed multipole analysis (DMA), pioneered by Stone [92,93]. Apart from this change, the approach was used as described in Ref. [51], to which we refer for further details. The transition densities required when computed used Multiwfn 3.7 [94], and the distributed multipole analysis was performed with the GDMA 2.3.3 program [95]. Preliminary tests using several ethylene and naphthalene dimers in the same configuration as were used in Ref. [51] were performed to ensure that the substitution of TrCAMM by DMA still leads to viable results; the differences in the values reported in Ref. [51] were smaller than 5%.

## 4. Conclusions

A significant contrast of the (hyper)polarizabilities has been observed between the *open–open, open–closed* and the *closed–closed* isomers of the studied compounds. NIBDT, as a linker, is associated with a greater contrast, in comparison to naphthalene, regarding the properties of interest. This has been attributed to the low-lying excited states of NiBDT. As it has been shown, the existence of low-lying excited states, due to the presence of NiBDT, significantly enhances the polarization character. 

The *closed–closed* isomer clearly demonstrates the great effect of conjugation on the (hyper)polarizabilities. The structural changes induced by photochromism can be closely followed by the second hyperpolarizability, and thus this property can be used as an index to monitor molecular changes, which are due to photochromism. A similar trend, although to a lesser extent, has also been found for the first hyperpolarizability. Overall, it is observed that NLO molecular properties can be used as probes for detection of the possible molecular states.

As expected, we found that the linker greatly affects the communication between the DTE groups, and thus, photochromism. It was found that a low V_DA_ is associated with NiBDT as a linker; therefore, both DTE groups are likely to close. The near-IR absorption spectrum of NiBDT could be associated with the observed low V_DA_ and diminished communication. We found that V_DA_ is also affected by molecular geometry. The computed low V_DA_ value for the case of the sexithiophene, R = Cl, molecular switch is in agreement with the experimentally observed full photochromism [37]. The linear response and the DMA methods give very close electrostatic contributions to V_DA_. 

Εnergy transfer may occur via the **oo → oc → cc** process. The vertical main emitting peaks of **oo** are very similar to the absorption peaks of **oc**. Similarly, the vertical main emitting peaks of **oc** and the adiabatic de-excitation energies are very similar to the absorption peaks of **cc**. 

Intramolecularly, the **cc** fragment presents major absorption peaks at ~650 nm, while the **oo** fragment does not present significant emission in this area. Thus, the bridge does not ease the charge transfer intramolecularly from the **o** fragment.

An overlap is observed between the fluorescence spectrum of **oc** with the absorption spectrum of **cc** and between the fluorescence spectrum of **oο** with the absorption spectrum of **οc**. The absorption and emission peaks at ~310 nm in all cases present charge transfer characters.

The results of the present study clearly reveal that the electronic nature of the NiBDT molecular bridge, connecting open–closed DTE units, could allow photoswitching between molecular states with tunable polarization characteristics. The sensitivity of the L&NLO properties could be used as a probe for the identification of the photoswitching process and the existence of the possible isomers.

## Figures and Tables

**Figure 1 molecules-28-05646-f001:**
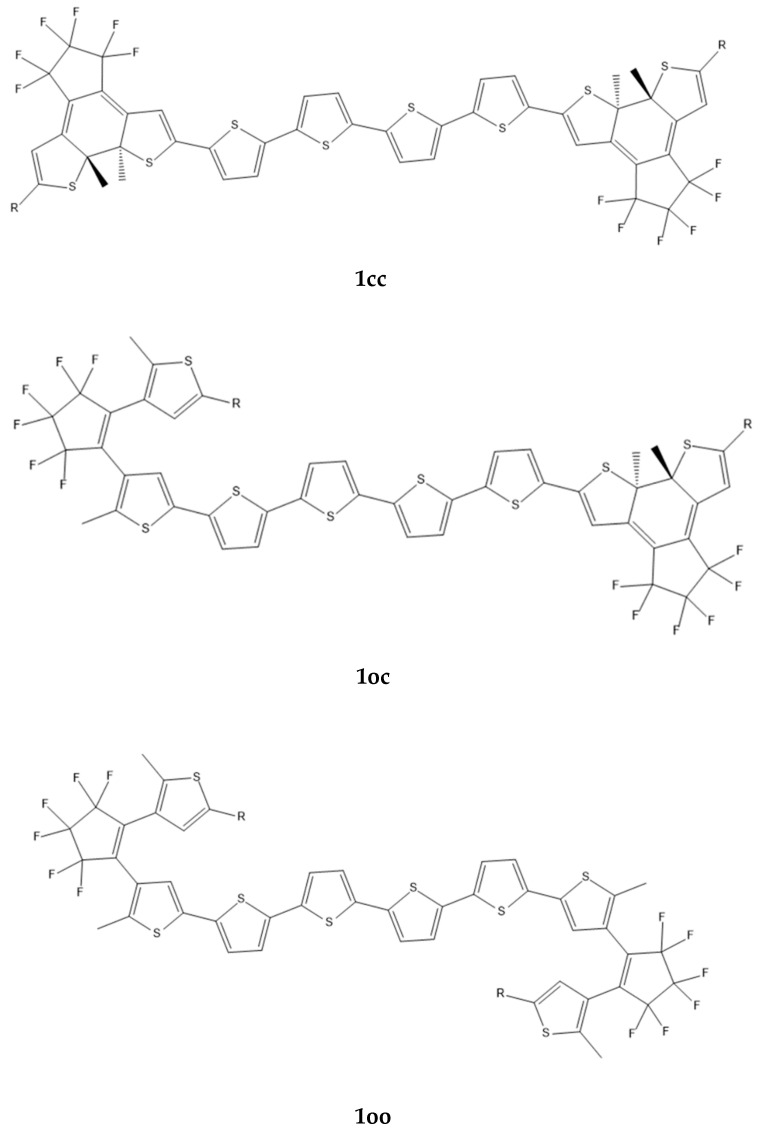
The structures of 1**cc**, 1**co**, and 1**oo** computed using the B3LYP/6-31G* method.

**Figure 2 molecules-28-05646-f002:**
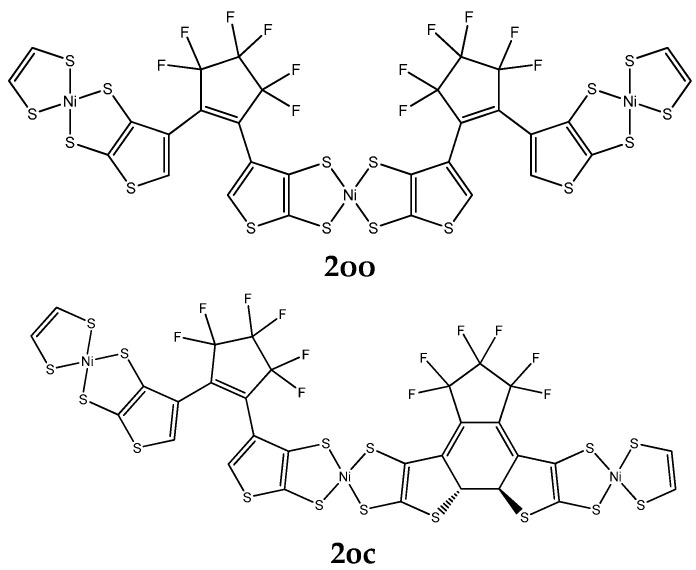
The structures of 2**oo**, 2**oc**, 2**cc,** 3**oo**, and 3**oc**, computed using the B3LYP/6-31G* method.

**Figure 3 molecules-28-05646-f003:**
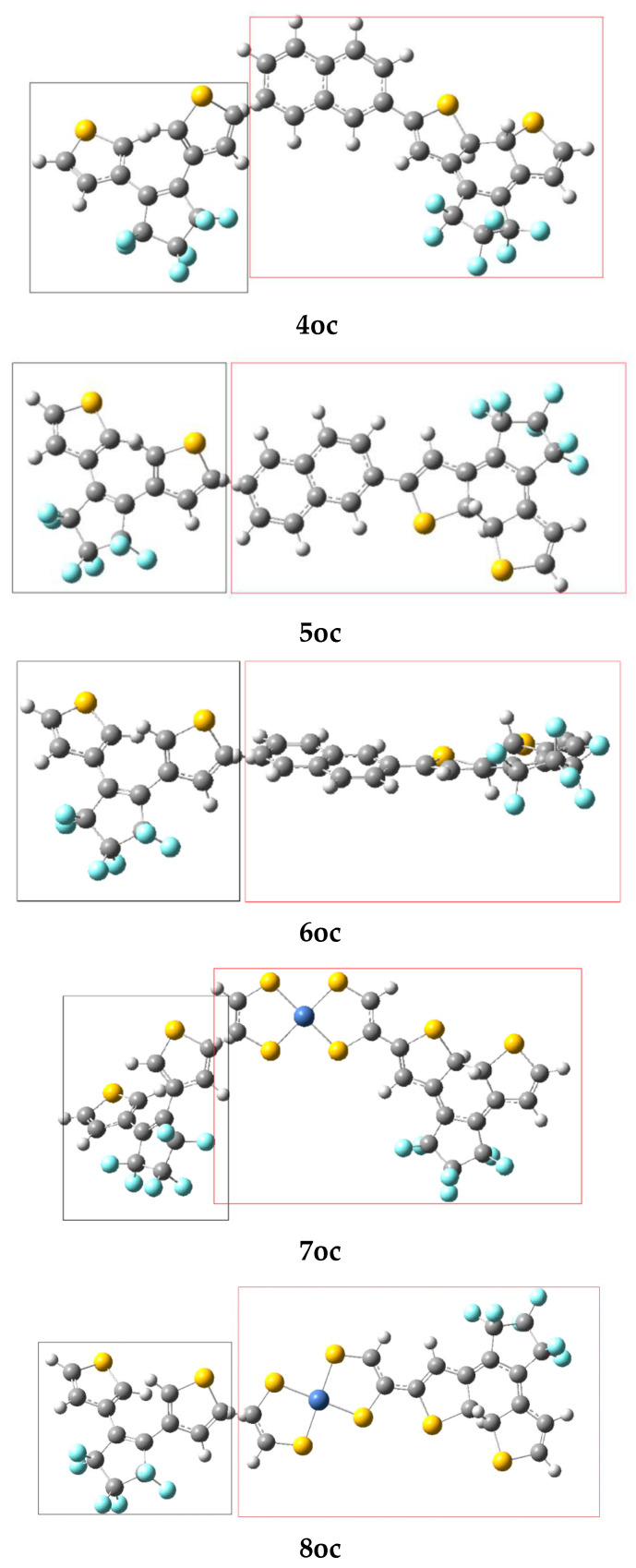
The considered dimers (left fragment: donor; right fragment: acceptor.

**Figure 4 molecules-28-05646-f004:**
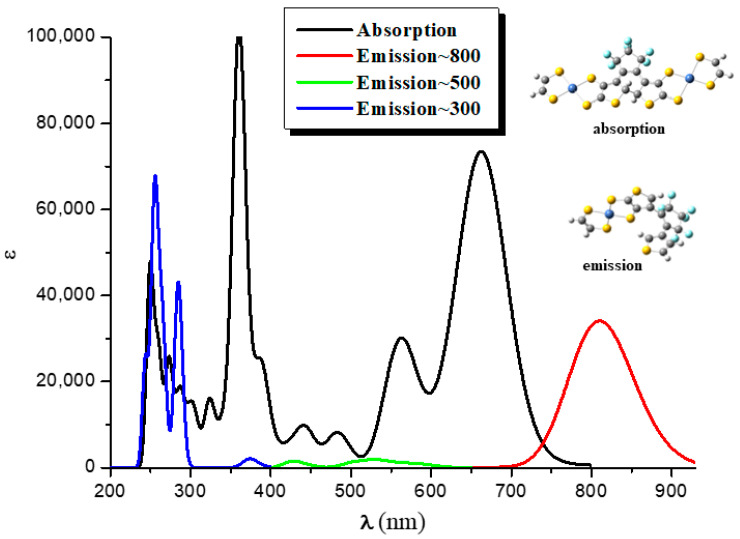
The absorption (black line) and the emission spectra (red, green, and blue lines) of the **2oc** fragments (left fragment: donor, right fragment: acceptor). The CAMB3LYP/6-31G* method was used.

**Figure 5 molecules-28-05646-f005:**
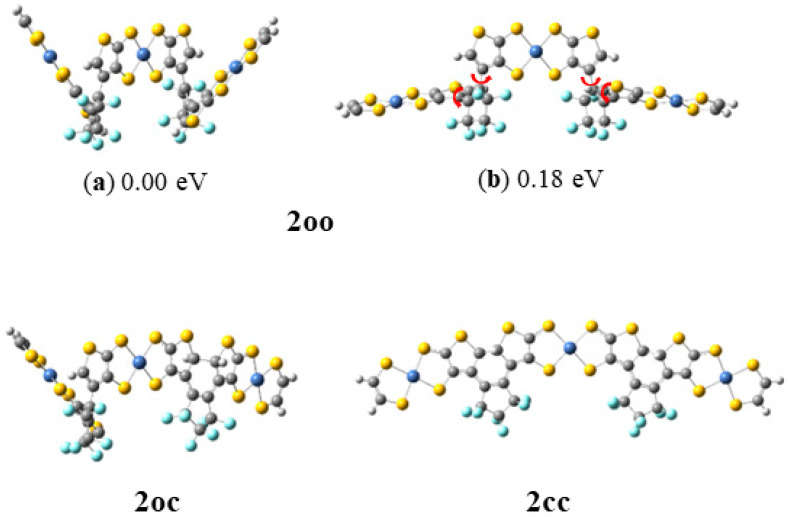
Calculated open–open (**2oo**), open–closed (**2oc**), and closed–closed (**2cc**) structures at CAM-B3LYP/6-31G* _H,C,O,F,S_ ECP28MWB(SDD)_Ni_.

**Figure 6 molecules-28-05646-f006:**
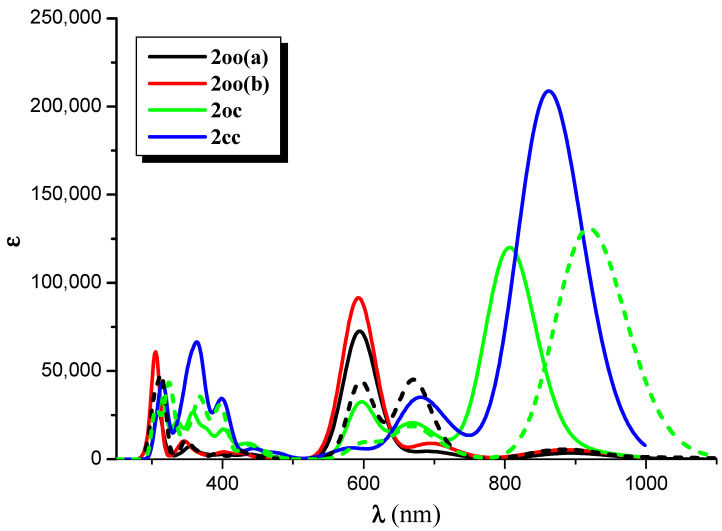
NIR-vis-UV absorption (solid line) and emission (dotted line) spectra of the **2oo**, **2oc**, and **2cc** structures at CAM-B3LYP/6-31G* _H,C,O,F,S_ ECP28MWB(SDD)_Ni_ level of theory. (Peak half-width at half height: 0.09 eV).

**Figure 7 molecules-28-05646-f007:**
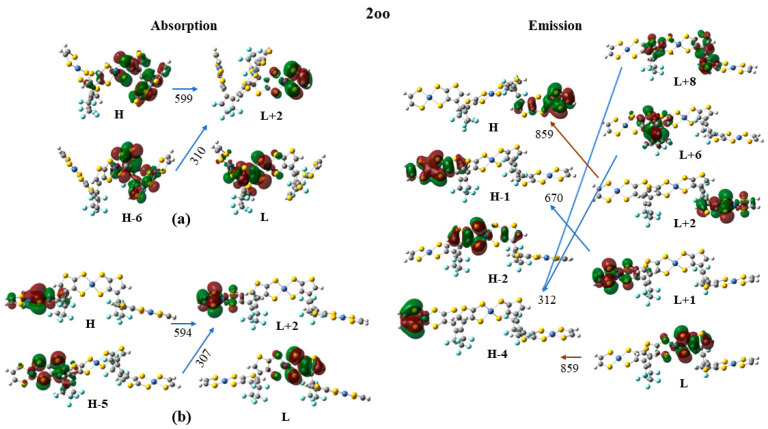
Frontier molecular orbitals (MO) involving in the main absorption and emission excitations of **2oo**. (**a**) absorption excitation of the **a** conformer (**b**) absorption excitation of the **b** conformer.

**Figure 8 molecules-28-05646-f008:**
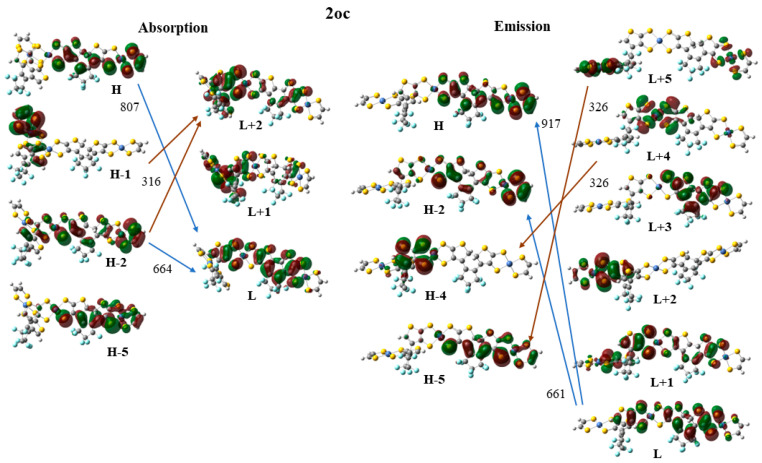
Frontier molecular orbitals (MO) involving in the main absorption and emission excitations of **2oc**.

**Figure 9 molecules-28-05646-f009:**
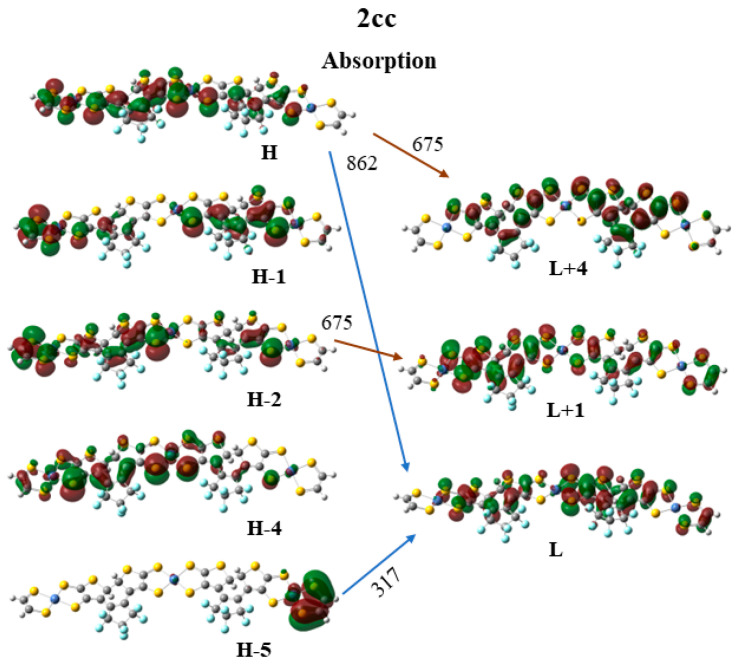
Frontier molecular orbitals (MO) involving in the main absorption excitations of **2cc**.

**Table 1 molecules-28-05646-t001:** The (hyper)polarizabilities of **1cc**, **1oc**, and **1oo** (Figure 1). The structures have been computed using the B3LYP/6-31G* method, and the (hyper)polarizabilities with the CAM-B3LYP/6-31G*. All values are given in a.u.

	α			β			γ (×10^3^)		
R	1cc	1co	1oo	1cc	1co	1oo	1cc	1co	1oo
H	972.6 1049.5 ^3^	882.3	796.9	680 790 ^3^	14,860	150	14,111 14,660 ^3^	8476	4078
Cl	1025.4	926.0	827.2	1200	20,860	243	15,635	9299	4185
	1108.0 ^3^			1250 ^3^			16,270 ^3^		
	914.0 ^4^	824.6 ^4^	776.7 ^4^	1960 ^4^	11,700 ^4^	144 ^4^	8654 ^4^	4719 ^4^	2837 ^4^
NO_2_	1123.8	988.8	842.5	257	48,040	773	22,841	13,260	4284
NH_2_	1024.4	918.9	822.2	530	401	486	16,212	8949	4208
Ph	1239.1	1096.1	960.8	−310	10,140	150	23,611	12,378	4692
NO_2_/ NH_2_ ^1^	1094	930.0	832.4	71,350	2950	4950	25,370	9049	4272
NO_2_/NH_2_ ^2^		988.0			61,260			14,861	

^1^ NO_2_/NH_2_ groups are anchored on open/closed unit. ^2^ NO_2_/NH_2_ groups are anchored on closed/open unit. ^3^ Basis set: C, S, F, Cl, H: cc-pVTZ. ^4^ Properties have been computed at the M062-X-optimized geometry.

**Table 2 molecules-28-05646-t002:** The absorption wavelength (λ; nm) of the lowest-lying allowed transition and the (hyper)polarizabilities (a.u.) of **2oo**, **2oc**, **2cc**, **3oo**, and **3cc** (Figure 2).

Derivative	λ	α	β	γ (×10^3^)
**2oo**	622.3 ^1^	880.3 ^1^	−410 ^1^	2741 ^1^
**2oc**	884.1 ^1^	1163 ^1^ 1225 ^2^	1349 ^1^ 1330 ^2^	17,400 ^1^ 19,236 ^2^
**2cc**	994.3 ^1^	1576.9 ^1^ 1652.5 ^2^	−14,076 ^1^ −13,080 ^2^	67,870 ^1^ 68,890 ^2^
**3oo**	614.0^1^	792.9 ^1^	432 ^1^	1416 ^1^
**3cc**	878.9 ^1^	1129.6 ^1^	5800 ^1^	13,971 ^1^

^1^ (U) CAMB3LYP/6-31G*. ^2^ Basis set: C, S, F, H: cc-pVTZ, Ni: SDD.

**Table 3 molecules-28-05646-t003:** Computed HOMA index of NO_2_ -1AB-NH_2_ and H-1AB-H, A,B = c,o.

	NO_2_ -1AB-NH_2_				H-1AB-H		
	cc	co	oc	oo	cc	oc	oo
I_HOMA_	0.827	0.806	0.807	0.778	0.843	0.812	0.790

**Table 4 molecules-28-05646-t004:** Electronic couplings (V_DA_) and different contributions (V_c_, V_xc_) for the considered dimers. All values are given in cm^−1^. All values were computed with CAM-B3LYP/6-31G* method.

Dimer	V_DA_	V_c_	V_xc_
**1oc ^a^**	8.3	8.1 9.7 ^b^	0.2 0.25 ^b^
**2oc**	10.3	9.7	0.6
**4oc**	101.5	101.6	−0.1
**5oc**	113.0	111.3	1.7
**6oc**	120.4	120.9	−0.5
**7oc**	32.3	30.9	1.4
**8oc**	12.8	12.1	0.7

^a^ R = Cl (Figure 1). ^b^ Properties have been computed at the M062-X-optimized geometry.

**Table 5 molecules-28-05646-t005:** Comparison of the Coulomb-term contribution to the total EET, computed using a multipole expansion of the transition density, with analytically computed values; all values are in cm^−1^ (~10^−4^ eV).

l + l′ Bridge	0	1	2	3	4	Analyt. Coulomb ^a^	Analyt. EET ^a.^
Naphthalene	82.3	137.1	143.6	127.4	117.8	111.3	112.9
NiBDT	4.8	8.1	13.7	12.1	15.3	12.1	12.9

^a^ The distributed multipole moments are computed analytically (using, in our case, Stone’s GDMA method) in contrast to the “fitting” procedures used by other approaches, which fit the multipole moments to the external charge density.

**Table 6 molecules-28-05646-t006:** Excitation energies, ΔE (eV), λ_max_ (nm), and f-values for the main peaks of the absorption and emission spectra and the corresponding main excitations of the **2oo**, **2oc**, and **2cc** structures at CAM-B3LYP/6-31G* (H,C,O,F,S) ECP28MWB(SDD)(Ni) level of theory.

Struct	ΔΕ	λ_max_	f	Excitations
**Absorption**				
**2oo(a)**	1.805	686.9	0.0115	0.35|H-2 → L + 1> − 0.29|H → L + 2 > + 0.28|H-1 → L + 1 > + 0.26|H-2 → L>
	2.071	598.7	0.1141	0.20|H → L + 2 > + 0.16|H-1 → L + 2 > − 0.21|H-10 → L + 5>
	4.006	309.5	0.1252	0.19|H-0 → L + 2 > − 0.13|H-1 → L + 6>
**2oo(b)**	1.791	692.2	0.0447	0.56|H-1 → L + 1 > − 0.36|H → L + 2>
	2.088	593.7	0.4031	0.42|H → L + 2 > + 0.23|H-1 → L + 1> − 0.18|H-2 → L>
	4.044	306.6	0.2014	0.18|H-2 → L + 8 > + 0.20|H-5 → L + 2 > − 0.22|H-35 → L> − 0.20|H-19 → L + 3>
**2oc**	1.537	806.9	0.7112	0.58|H → L > − 0.23|H-5 → L>
	1.868	663.7	0.0926	0.38|H-2 → L > − 0.27|H-5 → L>
	3.925	315.9	0.0983	0.39|H-17 → L > + 0.22|H-20 → L > + 0.31|H-20 → L + 2>
**2cc**	1.438	862.3	1.3931	0.51|H → L > + 0.30|H-1 → L + 1>
	1.837	675.1	0.1866	0.23|H → L + 4 > + 0.20|H-4 → L > − 0.23|H-2 → L + 1> + 0.18|H-1 → L>
	3.107	399.0	0.1077	0.28|H-1 → L + 1 > + 0.21|H → L + 4 > − 0.21|H-4 → L + 1> − 0.18|H-0 → L>
	3.369	368.0	0.1701	0.29|H-17 → L > − 0.16|H-18 → L > + 0.15|H-8 → L>
	3.519	352.4	0.1073	0.27|H-19 → L + 1 > + 0.19|H-19 → L > − 0.18|H-9 → L + 1> − 0.11|H-2 → L + 1>
	3.917	316.5	0.1402	0.27|H-10 → L + 1 > + 0.21|H-10 → L >− 0.19|H-13 → L> + 0.26|H-5 → L>
**Emission**				
**2oo**	1.444	858.7	0.0170	0.65|H-0 → L > − 0.15|H-2 → L>
	1.851	669.7	0.1938	0.47|H-1 → L + 2> − 0.51|H-1 → L + 1 > − 0.39|H-10 → L + 2 > + 0.27|H-12 → L + 2 > + 0.21|H-10 → L + 1>
	2.074	597.8	0.1588	0.52|H → L + 2 > + 0.40|H-9 → L + 4>
	3.966	312.6	0.2401	0.33|H-2 → L + 8 > + 0.14|H-2 → L + 6> − 0.36|H-30 → L>
**2oc**	1.353	916.5	0.7071	0.57|H → L > − 0.24|H-4 → L>
	1.874	661.6	0.0774	0.32|H-5 → L > + 0.29|H → L + 1 > − 0.43|H-2 → L> − 0.25|H → H-19>
	3.806	325.8	0.0748	0.31|H-15 → L > + 0.23|H-4 → L + 4 > − 0.25|H-5 → L + 5>

**Table 7 molecules-28-05646-t007:** Vertical and adiabatic excitation energies in eV of the main peaks of the **2oo**, **2oc**, and **2cc** structures at CAM-B3LYP/6-31G* (H,C,O,F,S), ECP28MWB(SDD)(Ni).

2oo	2oc	2cc	2oo	2oc	2oo	2oc
S_0_ → S_i_ ^a^			S_i_ → S_0_ ^b^		S_0_ → S_i_ ^c^	
1.81 (1.79) ^d^	1.54	1.44	1.44	1.35	1.67	1.40
2.07 (2.09) ^d^	1.87	1.84	1.85	1.86	1.98	1.87
4.01 (4.04) ^d^	3.92	3.11	3.97	3.81	4.00	3.88
		3.37				
		3.52				
		3.92				

^a^ Vertical absorption at the geometry of the ground state. ^b^ Vertical emission at the geometry of the emitted state. ^c^ Adiabatic excitation. ^d^ Main peaks of the b structure in parenthesis.

## Data Availability

Numerical data from computer simulations are available from the corresponding author upon request.

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
