# Peer review of "Photoswitchable Molecular Units with Tunable Nonlinear Optical Activity: A Theoretical Investigation"

_molecules, 2023, doi:10.3390/molecules28155646_

Round 1

Reviewer 1 Report

In the present manuscript entitled "Photoswitchable Molecular Units with Tunable Non-Linear Optical Activity: A Theoretical Investigation", the authors report a computational study on the molecular hyperpolarizabilities of a series of derivatives consisting of two dithienylethene (DTE) switches. They address the influence of substituents and linkers, and find that the NiBDT molecular bridge could result in a significant contrast between the hyperpolarizabilities of open and closed isomers. This effect is attributed to the low-lying excited states of NiBDT. They have also investigated the effect of molecular linker on the electronic communication between the DTE units and thus the photochromism. These studies indicate that the electronic nature of the NiBDT linker may enable us to access photoswitching between molecular states with tunable polarization characteristics.

This proposed article deserves to be published and the Molecules is certainly well targeted. Before the publication, I would like to ask the authors to consider the minor comments below.

1. It would be better to shorten the abstract. A good abstract needs to be concise and should avoid detailed descriptions.

2. A number of equations appear to be inserted by a direct cut-and-paste in terms of images, and they should be cleared as typesetting mathematics is straightforward and fairly easy using LaTeX.

3. Page 4, line 187

"...by employing the M062-X functional, since the latter has been shown to predict satisfactorily the geometries of π-conjugated systems, due to the appropriate amount (28%) of the included HF exchange."

To be more precise, the hybrid functional M06-2X includes 54% of the Hartree-Fock exchange.

4. Can the authors specify how they explored the structural changes due to photochromism?

Author Response

Reviewer #1
  1. “In the present manuscript entitled "Photoswitchable Molecular Units with Tunable Non-Linear Optical Activity: A Theoretical Investigation", the authors report a computational study on the molecular hyperpolarizabilities of a series of derivatives consisting of two dithienylethene (DTE) switches. They address the influence of substituents and linkers, and find that the NiBDT molecular bridge could result in a significant contrast between the hyperpolarizabilities of open and closed isomers. This effect is attributed to the low-lying excited states of NiBDT. They have also investigated the effect of molecular linker on the electronic communication between the DTE units and thus the photochromism. These studies indicate that the electronic nature of the NiBDT linker may enable us to access photoswitching between molecular states with tunable polarization characteristics.

This proposed article deserves to be published and the Molecules is certainly well targeted.

Authors' reply: We thank the referee for the careful reading of our manuscript and his/her positive evaluation of our research article.

  1. “This proposed article deserves to be published and the Molecules is certainly well targeted. Before the publication, I would like to ask the authors to consider the minor comments below.

 It would be better to shorten the abstract. A good abstract needs to be concise and should avoid detailed descriptions.

Authors' reply: We thank the reviewer for his/her suggestion. The abstract has been modified appropriately as follows.

Abstract: The first, second and third-order molecular nonlinear optical properties, including two-photon absorption of a series of derivatives, involving two dithi-enylethene (DTE), groups, connected by several molecular linkers (bis(ethylene-1,2-dithiolato)Ni- (NiBDT), naphthalene, quasilinear oligothiophene chains) are investigated by employing Density Functional Theory (DFT). These properties can be efficiently controlled by DTE switches, in connection with light of appropriate frequency. NiBDT, as a linker, is associated with a greater contrast, in comparison to naphthalene, between the first and second hyperpolarizabilities of the “open-open” and the “closed-closed” isomers. This is explained by invoking the low-lying excited states of NiBDT.   It is shown that the second hyperpolarizability can be used as the index, which follows the structural changes induced by photochromism. Assuming a Förster type transfer mechanism the intramolecular excited-state energy transfer (EET) mechanism is studied. Two important parameters related with this are computed: the electronic coupling (VDA) between the donor and acceptor fragments as well as the overlap between the absorption and emission spectra of the donor and acceptor groups. NiBDT as a linker is associated with a low VDA value. We found that VDA is affected by the molecular geometry. Our results predict that the linker strongly influences the communication between the open-closed DTE groups. The sensitivity of the molecular nonlinear optical properties could assist with the identification of the molecular isomers.”

  1. “. A number of equations appear to be inserted by a direct cut-and-paste in terms of images, and they should be cleared as typesetting mathematics is straightforward and fairly easy using LaTeX.”

Authors' reply: We thank the reviewer for the critical and constructive reading of our manuscript. In the revised text we formatted appropriately   eqs  2-5, 8

  1. Page 4, line 187

"...by employing the M062-X functional, since the latter has been shown to predict satisfactorily the geometries of π-conjugated systems, due to the appropriate amount (28%) of the included HF exchange." To be more precise, the hybrid functional M06-2X includes 54% of the Hartree-Fock exchange.

Authors' reply: We thank the referee for the careful reading of our manuscript. It has been corrected:”[…] latter has been shown to predict satisfactorily the geometries of π-conjugated systems, due to the appropriate amount (54%) of the included HF exchange

  1. Can the authors specify how they explored the structural changes due to photochromism?

Authors' reply: We thank the reviewer for his/her comment. As it is shown in tables 1 &2 upon photoswitching, the main structural  change, which is imposed is the closing of ring unit while the structure of the molecular  linker(bridge) remains unaltered (see for example Fig. 1S). The three types of isomers, OO, OC, CC differ in their conjugation length and their charge transfer character, when a donor/acceptor group is anchored on the DAE unit. It was found that: 1.) by increasing the conjugation length the polarizability and second hyperpolarizability significantly altered (p. 13, […]The great effect of conjugation on the hyperpolarizabilities is clearly demonstrated by the closed-closed isomers). In order to further quantify the effect of the conjugation length upon photoswitching, a HOMA analysis (Res & Discussions, p. 13) has also been employed. 2.) the effect of charge transfer is demonstrated by eqs  9 &10.

One of the main conclusions of our study is that the molecular L&NLO properties are very sensitive to the structural changes and so these properties can assist with the identification of the possible isomers.

Reviewer 2 Report

The authors report the theoretical non-linear activity of the series of derivatives with two dithienylethene (DTE), groups and it is connected by different molecular linkers. The studies have been performed satisfactorily. My opinion is that the manuscript is interesting, and the methodology and description of the results have been competently done. Therefore, I recommend the publication of this work after minor revisions as follows:

In my point of view

1 . An introduction and Results and Discussion section should be brief and concise. It covers all the important points

2. The manuscript is too long and should be moved some part from the Methods into the Supporting Information part.

3. The chemical diagram of compounds should be shown in the results section and discuss the structural difference. It is faster to understand the structure of compounds as well as easy to follow the results.

4. “1oo” label missing in Figure 1. It should be included.

5. Kindly format the reference in the main text, e.g. line no: 167, 119, 100 and etc.,

Kindly format the reference in the main text, e.g. line no: 167, 119, 100 and etc.,

Author Response

Reviewer #2

  1. The authors report the theoretical non-linear activity of the series of derivatives with two dithienylethene (DTE), groups and it is connected by different molecular linkers. The studies have been performed satisfactorily. My opinion is that the manuscript is interesting, and the methodology and description of the results have been competently done. Therefore, I recommend the publication of this work after minor revisions as follows:

Authors' reply: We thank the referee for the careful reading of our manuscript, his/her recognition of our research and his/her positive evaluation.

  1. In my point of view

    1 . An introduction and Results and Discussion section should be brief and concise. It covers all the important points

  1. The manuscript is too long and should be moved some part from the Methods into the Supporting Information part.?

Authors' reply: We thank the reviewer for his/her comment, giving us the chance to improve the flow of our manuscript, so that the concept of our study  be better clarified.  We have appropriately modified the following sections of our manuscript: Abstract, Introduction, Methods (removal some part of two-photon absorption ), Results and Discussion (removal of some tables and redeployment in the SI section, tables 4S-10S).

  1. 3. The chemical diagram of compounds should be shown in the results section and discuss the structural difference. It is faster to understand the structure of compounds as well as easy to follow the results..

Authors' reply:  It has been implemented (p. 8,11, Res & Discussion).

  1. “1oo” label missing in Figure 1. It should be included.

Authors' reply: We thank the referee for the careful reading of our manuscript. It has been implemented (p.8)

  1. Kindly format the reference in the main text, e.g. line no: 167, 119, 100 and etc.,

Authors' reply: We thank the referee for the careful reading of our manuscript. We have appropriately formatted the references.

Reviewer 3 Report

The authors of this paper conducted a detailed computational study on a series of molecular derivatives constructed from two dithienylethene (DTE) groups. These groups were connected by various types of molecular linkers, such as bis(ethylene-1,2-dithiolato)Ni- (NiBDT), naphthalene, and quasi-linear oligothiophene chains.

The focus of their investigation was the computation of the second- and third-order nonlinear optical properties, specifically the molecular hyperpolarizabilities of the designed/selected derivatives. They discovered that these properties could be effectively manipulated by DTE switches, when exposed to light of a suitable frequency.

The study is intriguing and the computations performed are challenging, being highly sensitive to the method used. The authors' extensive experience in this subject is a significant value-add, enhancing the reliability of the results.

However, the quality of the manuscript is somewhat lacking, likely due to issues during the conversion from a Word document to PDF. Nevertheless, the manuscript is still readable and no significant issues are present.

Comments:

1. The method used by the authors to compute the properties of interest is not clearly explained. In the "Methods" section, it states: "The finite field perturbation theory (FPT) has been used for the calculation of the static (hyper)polarizabilities." It is unclear whether they computed the entire tensors or some components. Polarizabilities and the first hyperpolarizabilities are typically accessible via the CPHF method. Why did the authors opt for a numerical method over an analytic approach?

2. Equation 10 is used to examine the charge transfer character on the second hyperpolarizability. However, the connection between this equation and the process is unclear. It would be beneficial if the authors could provide more details to clarify this connection.

The quality is fine. Minor issues will be taken care of during the final submission

Author Response

Reviewer #3

  1. The authors of this paper conducted a detailed computational study on a series of molecular derivatives constructed from two dithienylethene (DTE) groups. These groups were connected by various types of molecular linkers, such as bis(ethylene-1,2-dithiolato)Ni- (NiBDT), naphthalene, and quasi-linear oligothiophene chains.

The focus of their investigation was the computation of the second- and third-order nonlinear optical properties, specifically the molecular hyperpolarizabilities of the designed/selected derivatives. They discovered that these properties could be effectively manipulated by DTE switches, when exposed to light of a suitable frequency.

The study is intriguing and the computations performed are challenging, being highly sensitive to the method used. The authors' extensive experience in this subject is a significant value-add, enhancing the reliability of the results.

However, the quality of the manuscript is somewhat lacking, likely due to issues during the conversion from a Word document to PDF. Nevertheless, the manuscript is still readable and no significant issues are present.

Authors' reply:  We thank the referee for the careful reading of our manuscript, his/her recognition of our research and his/her positive evaluation. We greatly appreciated his/her comments.

  1. The method used by the authors to compute the properties of interest is not clearly explained. In the "Methods" section, it states: "The finite field perturbation theory (FPT) has been used for the calculation of the static (hyper)polarizabilities." It is unclear whether they computed the entire tensors or some components. Polarizabilities and the first hyperpolarizabilities are typically accessible via the CPHF method. Why did the authors opt for a numerical method over an analytic approach?

Authors' reply: In order to evaluate all the appropriated tensor components needed for the computation of the average values of the (hyper)polarizabilites (eqs. 2-5), the FPT approach was employed. The selection of the field step was safeguarded with the aid of Romberg-Rutishauer method. To make this issue clear, an appropriate definition was added (p.4) : “The finite field perturbation theory (FPT) has been used for the calculation of all the necessary tensor components of the static (hyper)polarizabilities

Why did the authors opt for a numerical method over an analytic approach?

Authors' reply: The targeted NLO property of our study is the second hyperpolariability. The software we used for the (hyper)polarizability computations, GAUSSIAN, is not able to calculate analytically  the second hyperpolarizability, γ. Other softwares (e.g GAMESS, DALTON) can compute the  analytic second hyperpolarizability, however it is not possible to apply a broken symmetry solution to the wavefunction, which efficiently takes into account static correlation effects and it is needed to be considered for all the isomers presented in  figure 2.

  1. Equation 10 is used to examine the charge transfer character on the second hyperpolarizability. However, the connection between this equation and the process is unclear. It would be beneficial if the authors could provide more details to clarify this connection.

Authors' reply:   Both Eq. 9 and 10 are only approximately valid as indices of the charge transfer effect on the hyperpolarizabilities. Eq. 10 can be shown to be valid under the two-state approximation, if transition moments and frequencies of the three species are approximately equal. We added the following sentence after Eq. 10: “This relationship can be shown to be approximately valid in the two-state approximation to γ [89], if the average transition dipole of the two symmetrically substituted compounds is similar to the transition moment of the unsymmetrically substituted one, and the transition frequencies are also similar.” A new reference was also added.